# Gut Microbiome Proteomics in Food Allergies

**DOI:** 10.3390/ijms24032234

**Published:** 2023-01-23

**Authors:** Ana G. Abril, Mónica Carrera, Ángeles Sánchez-Pérez, Tomás G. Villa

**Affiliations:** 1Department of Microbiology and Parasitology, Faculty of Pharmacy, University of Santiago de Compostela, 15706 Santiago de Compostela, Spain; 2Department of Food Technology, Spanish National Research Council, Marine Research Institute, 36208 Vigo, Spain; 3Sydney School of Veterinary Science, Faculty of Science, University of Sydney, Sydney, NSW 2006, Australia

**Keywords:** gastrointestinal track, microbiota, mass spectrometry, metaproteomics, proteomics, food allergies

## Abstract

Food allergies (FA) have dramatically increased in recent years, particularly in developed countries. It is currently well-established that food tolerance requires the strict maintenance of a specific microbial consortium in the gastrointestinal (GI) tract microbiome as alterations in the gut microbiota can lead to dysbiosis, causing inflammation and pathogenic intestinal conditions that result in the development of FA. Although there is currently not enough knowledge to fully understand how the interactions between gut microbiota, host responses and the environment cause food allergies, recent advances in ‘-omics’ technologies (i.e., proteomics, genomics, metabolomics) and in approaches involving systems biology suggest future headways that would finally allow the scientific understanding of the relationship between gut microbiome and FA. This review summarizes the current knowledge in the field of FA and insights into the future advances that will be achieved by applying proteomic techniques to study the GI tract microbiome in the field of FA and their medical treatment. Metaproteomics, a proteomics experimental approach of great interest in the study of GI tract microbiota, aims to analyze and identify all the proteins in complex environmental microbial communities; with shotgun proteomics, which uses liquid chromatography (LC) for separation and tandem mass spectrometry (MS/MS) for analysis, as it is the most promising technique in this field.

## 1. Imbalances in the Human GI Tract Microbial Ecosystem and Their Association with Food Allergies

The gastrointestinal (GI) tract constitutes one of the most complex microbial ecosystems, with thousands of bacterial species coexisting (or clashing) with protists, fungi and viruses (including bacteriophages). In humans, the GI tract contains an estimated of 10^11^ bacterial cells/mL [1,2]. Many of the microorganisms that inhabit the GI tract perform a role, either directly or indirectly, in the development of a variety of allergies which can affect as many as 4% of the adult human population. An assortment of factors, ranging from the manner of birth to the use of antibiotics, are known to produce dysbiosis in the gastrointestinal microbiota, hence modulation of the gut flora appears to be a good strategy to prevent or treat these allergies [3,4]. The gut microbiome is continuously evolving, from the instance of birth to the moment of death, with many factors playing a role in these changes. The development of food allergies (FA) is affected by diet, interactions with other people or household pets and medical treatments, such as the intake of gastric inhibitors and antibiotic therapies. The role of the GI tract microbiota in FA is currently being deciphered, but it is clear that beneficial commensal bacteria, essential for the maintenance of food tolerance, are negatively affected by the factors mentioned above [5]. Although more than 170 foods have been identified as being potentially allergenic, a minority of these foods cause the majority of reactions and common food allergens vary between geographic regions [6]. Moreover, the gut microbiomes of human populations worldwide have many core microbial species in common; however, within a species, some strains can show remarkable population specificity [7]. In addition, there exists both large interindividual and intraindividual variation in gut microbiota composition, which complicates the identification of microbial signatures associating with allergies. The reduction in intraindividual variation could be achieved by maintaining a steady lifestyle, including dietary habits among others, within the timeframe of the intervention study [8].

The post-natal gut microbiome, in infants, is particularly rich in bacteria belonging to the genera *Bifidobacterium* and *Lactobacillus*, these microorganisms are believed to originate a healthy immune regulatory response, positively related by gut T effectors (helper T cells) and secretory IgAs (Immunoglobulins A), the main antibody present in breast milk, and transcriptional regulators involved in gut homeostasis. The introduction of solid food into the diet, represents a new influx of microbiota, including bacteria belonging to the orders Clostridiales and Bacteroidetes. These bacteria were reported as involved in the suppression of IgE (Immunoglobulin E) responses, that cause FA [9]. The intestinal mucosa, the interface between the body and microbiota, was estimated to have a surface area of 250 m^2^. The mucosa is not only the absorptive area of the GI, allowing nutrient transport into either the circulatory or lymphatic systems, but also the entry point for bacterial agonists and pathological and virulence effectors; the latter must be suppressed by the beneficial commensal microbiota if intestinal homeostasis is to be maintained [10].

The development of IgE-mediated FA in infants, in particular egg intolerance, appear to be associated with the type of delivery, either vaginal or cesarean, although not all authors agree. It is generally accepted that cesarean delivery increase the risk of development sensitization to certain food allergens in children [5]. Intestinal dysbiosis can cause GI tract malfunction that, in serious cases, results in a breach in the intestinal barrier; ‘leaky gut syndrome’ is a disease in which the abnormal intestinal permeability results in gut antigens reaching the blood stream and originating allergic reactions in other organs, such as the lungs in atopic asthma [2]. It is currently not possible for many of gut microorganisms to be cultured axenically in the laboratory, either for lack of the appropriate culture media or insufficient knowledge. hHence, a novel technique, known as shotgun metagenomic sequencing, is invaluable to assess the role of the different members of the microbiota, bacteria, fungi, protists or viruses on dysbiosis; this method involves untargeted (shotgun) sequencing of all (meta = ‘transcendent’) the microbial genomes (genomics) present in a complex sample. Although there is still not enough data available to fully comprehend the dynamics of the intestinal gut microbiota, a better understanding of the GI tract colonizers will, undoubtedly, bring an array of microbial therapeutics to treat current GI tract dysbiosis and improve human health [11].

During their first year of life, babies can suffer from FA mediated by IgA, due to low numbers of Clostridiales in their guts, as compared to their healthy counterparts [12]. Apart from this microbial group, other bacteria, such as *Leuconostoc*, *Weissella* and *Veillonella*, are also found in low numbers in babies suffering from FA. On the other hand, particular bacteria appear to flourish under these conditions, which include microbes belonging to Enterobacteriaceae and Parabacteroides. These FA can also be accompanied by a reduction in the lactate-utilizing bacteria, hence resulting in lacto-dysbiosis. There are a number of bacterial taxons that appear to be involved in the preservation of GI tract homeostasis, with Lachnospiraceae and Ruminococcaceae as prime examples [13]. The authors conducted long term studies in 18 pairs of twins, ranging in age from 6 months to 58 years. In 13 of these sets of twins, one individual suffered from a food allergy while the second twin was healthy, whilst in the remaining 5 pairs, both twins experienced FA. As expected, the fecal samples analyzed in this study revealed mayor differences between the twins suffering from FA and their healthy counterparts; the study identified differences in bacterial composition, in species belonging to 68 microbial taxa in the twins affected by FA, as compared to their healthy twins, demonstrating that certain microbial families exert a protective role in the GI, preventing the development of FA, including those against milk [13]. An additional conclusion of the study was that certain metabolites produced by commensal bacteria, such as diacylglycerol, were found to be present at higher concentrations in healthy twins than in their allergic counterparts. Bao and colleagues reported that *Phascolarctobacterium faecium* was the main GI tract microorganism responsible for diacylglycerol production, with the authors identifying another member of the intestinal microbiota, *Ruminococcus bromii*, as involved in both starch digestion and the metabolism of amino acids and sterol [13]. Table 1 summarizes relevant alterations in the composition of the gut microbiome in individuals suffering from particular FA as compared with healthy subjects. The use of germ-free animals, completely devoid of microbes, also constitutes an important tool to study the role of microorganisms in the onset of allergies triggered by food [14]; it was demonstrated that fecal microbiota from allergy-suffering animals, when implanted into germ-free organisms, resulted in the recipient animal developing the same FA as those displayed by the donor. Even more remarkable, previous inoculation of the germ-free animals with a single bacterial species (i.e., *Anaerostipes caccae*), prevented the animal from developing FA.

The review by [25] includes a variety of studies, in both human and mice, which confirm the usefulness of fecal transplants, containing beneficial bacteria, in the treatment of patients suffering from FA. This therapeutic approach has also produced positive outcomes in the treatment of autoimmune diseases, such as Crohn disease and ulcerative colitis, triggered by bacteria present in the GI. In particular, it involves the restoration of the retinoic orphan receptor ϒ T (RORγt)+ regulatory T cells, by a mechanism dependent of MyD88 (Myeloid differentiation primary response 88), that allow the production of healthy immunoglobulin A [12,26]. Both gut microbiota secondary metabolites (such as short chain fatty acids) and complex surface microbial molecules with immunomodulatory activity are involved in the enhancement of RORγt+ Treg cell population differentiation [9,27].

## 2. Allergic Reactions in the GI

As it is well known, the plasma cells present in the intestinal lamina propria (the layer of connective tissue located under the mucosal epithelium), synthesize and secret the majority (*circa* 80%) of the antibodies secreted by healthy human beings; these include immunoglobulins G, M, A, D and E, although IgM and IgA are the key players in the GI tract. These antibodies are transported from the lamina propria via the polymeric immunoglobulin receptor protein (pIgR) which, as a transmembrane protein, facilitates immunoglobulin secretion across the intestinal epithelium; the second step involves the receptor protein being hydrolyzed by proteases, thus releasing the immunoglobulins into the intestinal lumen, together with the remainder of pIgR, known as secretory component (SC). The secreted immunoglobulins (mostly IgA) engage in a variety of intestinal processes, contributing to intestinal homeostasis, and playing a major role in the development of a healthy gut microbiota [10]. In addition, the regulatory T cells (Tregs) are also involved in GI tract homeostasis by controlling the inflammation processes since the absence of Tregs would escalate into an autoimmune pathological disorder l [28,29,30]. Upon induction by the intestinal microbiota, the regulatory T cells secrete a transcription factor known as ROR-ϒt (Retinoid –related Orphan Receptor ϒt), involved in controlling the activation of the helper cells Th17, hence, performing a crucial role in the prevention of pathological conditions mediated by Th2 cells. This situation can gather momentum as the ROR-ϒt declines and new Th2 cells are generated. Th2 produce a variety of cytokines, such as interleukins IL-13, IL-10, IL-4 and IL-5, which exert major immunomodulatory activities, including promoting B cells to synthesize high levels of cell-surface IgE [31]. As depicted in Figure 1, IL-17, produced by Th 17 cells, is a strong inflammatory cytokine involved in allergen sensitization [29]. Noval et al. (2015) [32] demonstrated the association of FA with the presence of both Th2 cells and allergen-specific high affinity IgE antibodies; however, the development of oral tolerance to allergens present in foods is controlled by Tregs that express Foxp3 (forkhead box P3) and are essential cells to maintain immune tolerance.

The pathologic events in FA involve, as mentioned above, activation of Tregs, that synthesize transcription factors, such as GATA3 (GATA sequence binding protein 3) and IRF4 (interferon regulatory factor 4), and the synthesis of high levels of interleukins IL-4 and IL-3. IL-4 activates STAT6 (Signal Transducer Activator transcription 6), which involves a pathway that represses the production of TGF-β1 (Transforming Growth Factor β1) by Treg cells, which concentration is decreased in patients affected by FA, as compared to healthy people. Hence, the food allergy originates from the failure of the immune system to control the population of mast cells. IgEs binds with high affinity to the Fc receptor (FcRI) present on the surface of mast cells [9], triggering mast cell degranulation and releasing multiple pro-inflammatory mediators, degranulation increases cell permeability and produces an allergic reaction that, if severe, can result in anaphylaxis, a systemic and potentially fatal immune response. A subset of TfH (T follicular helper) cells, known as Tfh13, were recently described to synthesize high amounts of IL-21, IL-13 and IL-4; these cells modulate the production of high-affinity IgE, at least in mouse models of the disease [9]. The presence of these cells, in conjunction with the reduced production of TGF-β1 by Treg cells, results in a reduced tolerance to food allergens [33]. Some metabolites, although produced by ‘healthy’ GI tract microorganisms, can affect the survival of other microbial colonizers (including pathogenic organisms) and can even be transported into the lamina propria. Such compounds, present both in mice and humans, include tryptophan derivatives, short-chain fatty acids and secondary bile acids; these metabolites are involved both in the modulation of IgA production and in the general intestinal humoral immune response. In fact, patients suffering from inflammatory bowel disease display increased levels of IgAs and IgGs; IgAs attach to bacteria and these cell-surface IgAs (sIgA) can be transported into the mucosa via microfold (M) cells. These are specialized epithelial cells, within the intestinal mucosa, which can capture luminal antigens and deliver them, by transcytosis, to the immune cells located beneath them. The antigen-bound IgGs can also enter the lamina propria by transcytosis and, once there, they bind to the receptor FcϒR (glycoproteins that recognize the Fc region or tail of IgGs) and activate the macrophage inflammasome, thus escalating inflammation [1,9,10].

IgAs are the main immunoglobulins in the gut and are involved in generating the necessary microbial diversity to develop GI tract homeostasis. This homeostasis can be disrupted by a number of factors, including dysregulation of the IgA response and colonization by pathogens that induce FA. One such example is *Schistosoma haematobium*, a trematode associated with the production of IgEs that cross-react with epitopes present in peanuts; hence, people infected with this parasite develop an allergy to this nut. On the other hand, the intestinal microbiota can positively reinforce the oral tolerance to certain food allergens. The mechanism involves the induction of IL-22 innate lymphoid cells type 3 (ILC3) that, apart from performing a role as antigen-presenting cells, induce a reduction in mucosal permeability and promotes mucus secretion by goblet cells, two actions that prevent seepage of protease-resistant allergens through the intestinal barrier. The innate lymphoid cells 3 (ILC3) are involved in the production of IL-2 and absence of this interleukin correlates with lower levels of gut Tregs cells and, therefore, an increase in the tolerance of food antigens [9]. As mentioned above, the transcription factor RORϒt (retinoic acid-related orphan receptor gamma t) may be induced by particular commensals bacteria, similar to those belonging to the orders Clostridiales and Bacteroidales, but only if nascent Tregs cells are present; this induction involves a MyD88-dependent mechanism, and increase tolerance to food antigens [1,34,35]. In fact, a few Bacteroidales species have already been successfully used as therapeutic agents to combat FA. Moreover, even treatment with a single species, *Subdoligranulum variable* (a Gram-negative, non-spore-forming, butyrate-producing anaerobic bacterium), related to the *Clostridium leptum* supra-generic rRNA cluster [36] also reduces FA and involves the mechanism described above. This is also the case for *Anaerostipes caccae* (an anaerobe, butyrate-producing, Gram-variable bacterium), reported to lessen the allergic reactions in mice sensitive to β-lactoglobulin. Food allergy sensitive mice (Il4raF^709^; C.129X1-Il4ratm3.1Tch), which exhibit a functional mutation in the Interleukin-4 receptor-alpha chain (IL-4Ra) were studied by Noval and co-workers (2013) [37] in an experiment designed to elucidate the involvement of group 2 innate lymphoid cells (ILC2s) in FA; again, these experiments demonstrated that the susceptibility to FA is associated with changes in the intestinal microbiota [1,38]. These Il4raF^709^ mice represent a good animal model to study FA. One study, involving the sensitization of mice with OVA (ovalbumin) and SEB (staphylococcal enterotoxin B), followed by OVA challenge, demonstrated a significant reduction in food allergy in animals that received a fecal matter transplant from the wild type mice [12]. These results, taken together with those mentioned above for *S. variabile*, are consistent with the idea that MYD88 and the transcription factor RORϒt, present in natural Treg cells (nTregs) and involved in the control of inflammation, can, in fact, restore immune tolerance to food allergens [1,37]. On the other hand, GI tract commensal bacteria can also control, and perhaps even suppress, the abnormal intestinal bacteria colonizing the Il4raF^709^ mice suffering from FA. Indeed, human Bacteroidales (*Bacteroides fragilis*, *Bacteroides ovatus*, *Bacteroides vulgatus*, *Prevotella melaninogenica* and *Parabacteroides distasonis*) were also reported to promote oral tolerance of FA in this mouse model [12]. In addition, commensal bacteria are also believed to be dependent on ROR-γt+ Tregs to carry out their protective role against FA; this was demonstrated in Foxp3YFPCreRorc∆/∆ mice, a mouse model lacking ROR-γt expression [12]. Finally, there are also food supplements, used as microbial therapeutics, that can ameliorate or even cure FA by restoring the intestinal microbial homeostasis [5,39,40]; these include probiotics (live microorganisms that represent healthy gut bacteria) and synbiotics [41,42], the latter is a combination of probiotics and prebiotics (digestible fiber that helps healthy bacteria in the gut).

## 3. Proteomics to Study Food Allergies

The constant increase in the prevalence of FA urgently demands the development of novel methods for the prevention, improved diagnosis and treatment of this condition. The traditional biological techniques are inadequate and need to be replaced with sophisticated high-throughput approaches. The current methods used by allergists and immunologists involve (i) skin prick and intradermal tests; (ii) pulmonary function tests; (iii) determination of serum allergen-specific IgE levels, ranging from low to medium or high; and (iv) a variety of immunological tests, including determining the levels of immunoglobulin, tryptase (released from mast cells) and complement in blood and flow cytometry techniques to analyze basophil activation. However, all those techniques have many drawbacks and they are inadequate for diagnosis in some cases due to specificity and sensitivity limits; in addition, allergic responses vary considerably in different diseases, as they are heterogeneous in their pathophysiology phenotypes [43,44,45]. New omics scientific technologies allow a wider and more systematic investigation of a variety of biologically pertinent compounds; these novel approaches use high-throughput techniques, to characterize a set of molecules that are relevant in a particular field of biology, hence, stablishing their connection and interactions. These technologies include genomics, epigenomics, transcriptomics, proteomics, metabolomics, microbiomics and exposomics and allow the design of biological pathway models and unraveling complex regulatory networks. In addition, new advances in the field of bioinformatics permit the integration, analysis and interpretation of the information obtained, paving the way to the development of personalized medical treatments for FA [46]. The efficiency, and amelioration of the price, due to improvements in the acquisition of -omics datasets and the development of advanced analytic techniques to process the high amount of data produced, has resulted in an increase in the use of these techniques to study a variety of human diseases, including asthma, allergic rhinitis, FA and immunodeficiencies [43].

The term proteome encompasses all the proteins that constitute functional components of the cells and catalyze vital processes. Consequently, proteomics is the comprehensive study of not only the structure of these proteins, but also their functions and their interactions. Indeed, the purpose of discovery proteomics is the identification of the individual proteins that constitute the proteome, changes in their cellular levels, characterization of the post-translational modifications undergone by the polypeptides and recognition of protein/peptide biomarkers [47]. Food proteomics represents one of the main tools to study the peculiarities and nature of food allergens [48]. In fact, the application of proteomics to food allergen identification, characterization and quantification is known as allergenomics [49], with the proteins involved in allergic responses constituting the allergenome. There are currently a variety of databases that compile the nomenclature and structure of food and environmental allergenic proteins, which include: http://www.allergenonline.org/, http://www.allergen.org/, https://fermi.utmb.edu/ and http://www.allergome.org/, accessed on 18 December 2022.

Proteomic approaches to study food allergens can be divided into two groups: gel-based or gel-free. The gel-based workflow involves the use of 2D electrophoresis and 2D immunoblotting, followed by identification of the resulting polypeptides by MS. On the other hand, gel-free allergenomics is characterized by the use of techniques, such as HPLC-MS/MS and IgE-binding assay on the trypsinized proteome. In addition, bioinformatics tolls are required for the study of specific immunoreactive epitopes [48].

Allergenomics has so far permitted the identification of the primary structure of more than 850 allergens [50], including food allergens, with Carrera and coworkers reporting in 2010 that 25 novel beta-parvalbumins as fish allergens, the proteins were sequenced by the exclusive use of MS-based techniques [51]. Additional allergens were identified from different food sources, such as rice [52], rice endosperms [53], legumes (peanuts, soybeans and lentils) [54], wheat [55], fruit [56], hazelnuts, pistachio nuts and sesame seeds [57]. Some studies focused on allergens found in other types of food, such as in the proteins present in milk [58,59,60], baked milk products [61] and beef [62], and in beer [63], fish [64] and shrimp [28]. An interesting report concentrated on the digestion products of major peanut allergens to examine the IgE reactivity [65]. The techniques used for allergen identification varied according to the investigators. Carrera et al. identified the B-cell epitopes of fish allergens by shotgun proteomics [66], while Ortea et al. used LC-MS/MS DDA to characterize shrimp arginine kinase as allergen [67]. In addition, several reports in the literature used targeted proteomics to identify allergenic substances in other types of food [68], such as gluten [69,70], eggs [71,72], wine [73,74,75], shrimp [76], processed food [77], legumes [78], peanuts [79,80,81,82,83,84,85,86] and milk [72,87,88,89,90,91]. Houston et al. [92] analyzed commercial soybean varieties, using a label-free proteomics approach to evaluate the concentration of 10 allergens; while Koeberl and colleagues utilized the MRM (Multiple Reaction Monitoring) mode of the MS technique for targeted quantification to develop a method for allergen quantification [93]. There are also a variety of studies to handle identification of allergens in complex food products, that include the presence of whey allergen in fruit juices [94] and fish [95,96]. *Anisakis simplex* and *Anisakis pegreffii* are parasitic nematodes of marine species, and their oral intake can produce allergic reactions characterized by urticaria, angioedema, asthma, conjunctivitis, and, in extreme cases, anaphylactic shock. Protein extracts of *Anisakis*-like allergens were studied using targeted mass spectrometry analysis and immunological methods [97,98]. 

Proteomic analyses also permitted the identification of different disease subtypes in patients suffering from chronic rhinosinusitis [99,100] and asthma [101]. In addition, proteomics can also study cross-reactivities between allergens; matrix-assisted laser desorption ionization-time of flight (MALDI-TOF) mass spectrometry (MS) successfully identified as many as four peptides from α-casein (a cow’s milk allergen) and three peptides from Gly m 5 (an allergen present in soy milk) that share a common core motif, suggesting the possibility that these shared epitopes are responsible for the allergic cross-reactions between cow’s and soy milks [102]. On the other hand, proteomics also constitutes a useful tool to elucidate the host response to FA, in order to understand the integration of proteins and their involvement in the reactions. One such example is the characterization of bovine α-S1-casein, present in human colostrum, as the putative cause of allergic sensitization to cow’s milk in children that are breast-fed only [103]. In addition, recent technical advances, such as single-cell proteomics, allow quantitation of molecules at the individual cell level, providing a deeper understanding of key events that can only be observed at the cellular level as it can be masked by other biological processes [46]. These single-cell focused MS-based approaches were used, in combination with fluorescence-based flow cytometry and chelated antibody tags, labelled with isotopes not normally found in biological systems, to simultaneously measure 34 parameters in each individual cell. These type of studies permitted not only the identification of cell subset-specific signaling phenotypes, but also system-wide signaling profiles in human hematopoiesis and successfully evaluating the proteins present in serum [104,105,106,107]. In fact, conventional fluorescent flow cytometry can concurrently analyze more than 18 target proteins in a single cell, while cytometry by time-of-flight (CyTOF) is an improved technique that allows simultaneous analysis of up to 40 proteins (although, in theory, the number can be extended to 100). The above approach was also employed for the determination of allergen specific T cells by following their fate during the immunotherapy against particular FA [108,109] and the epitope repertoire of T-cells [110]. It also successfully recognized lymphoma neoantigens [40]; neoantigens are novel, altered proteins that originate from cancer cell-specific DNA mutations. Additional studies include analyzing the specific alterations, in a variety of organs, as a result of the allergic response; in particular, the spleen and intestine of mice sensitized with either high (shrimp and clam) or weak (fish) allergenic tropomyosins (protein that binds actin filaments and regulates muscle contractions). They have used positively the sequential windowed acquisition of all theoretical fragment ion mass spectra (SWATH-MS)-based proteomics. The results demonstrated that both Th1 and Th2 cells can be used as biomarkers for tropomyosin allergy in mice [111]. Additional technological advances reported include the combination of a series of techniques, nanodroplet sample preparation, ultra-low-flow (nano-LC), high-field asymmetric ion mobility spectrometry (FAIMS) and Orbitrap Eclipse, the latest-generation Tribrid mass spectrometer, for greatly improved single-cell proteome profiling [112]. However, there are yet no reports on the use of these novel platforms in the field of allergies. 

Data-independent acquisition coupled with ion mobility mass spectrometry–mass spectrometry (DIA–IM–MS) was used to study the allergen composition of both raw peanuts and roasted peanut flour ingredients [86]; quantification of 123 proteins indicated that Ara h 1 and 3, two allergens belonging to the cupin superfamily of proteins, were the most abundant and present in similar amounts; however, only reduced amounts of the polypeptides were extracted from roasted peanut flour. On the other hand, the polypeptides Ara h2 and 6, corresponding to the prolamins group of plant storage proteins, were less abundant, but not affected by roasting. The authors also reported that removing fat from the peanuts reduced the content of the oleosins Ara h a10 and Ara h 11, which is not surprising as these proteins are associated with plant seed oil bodies, including peanuts. Gluten peptides, from different types of wheat flour, were evaluated and relatively quantified using a non-targeted multiplexed MS-MS method with a high definition mass spectrometer equipped with a travelling-wave ion mobility (TWIM) separation device and hybrid analyzers (such as a quadrupole/ion mobility mass spectrometry/orthogonal acceleration time-of-flight (Qq-IMMS-*oa*TOF) geometry). Of the peptides identified by this approach, 19% displayed an immunogenic prolamin-associated reaction, whereas 60% corresponded to high-molecular-weight glutenins [113], storage proteins that constitute one of the components of gluten.

The recent technology revolution provides a wide range of technical advances, which include the design of miniaturized and portable medical instruments that can be used to identify a great variety of molecules, such as gluten, drugs, alcohol, sleeping pills, stimulants, molds and volatile substances. Nevertheless, the development of such devices must be accompanied with advances in software designed for data collection, storage and visualization. Blockchain-based technology is an advanced database that allows the storage, management and access of data to be evaluated; one such example is the FOODALERT platform, that combines sensors and potentiostats (devices that measure and control electrode potential) that permits the specific detection of gluten in food [114].

Nanosensors can be used for the detection of biomolecules, such as allergens [115]. Some sensor technology is based on fluorescence and uses aptamers (single-stranded nucleic acid sequences that specifically recognize and bind particular targets) for biorecognition. These aptasensors represent a major improvement in the field, as they are fast, specific and highly sensitive and fast and simple to use, as demonstrated for analysis of food allergens [116,117,118,119,120,121,122,123,124]. The incorporation of a fluorescent tag into biosensors also provides additional advantages, ranging from not needing to use radioactivity to their simplicity of use, sensitivity and potential for high throughput screening.

### 3.1. Allergy Proteomics and Microorganism

Different -omics techniques, including proteomics, are currently used to analyze the multifaceted relationship between composition of the GI tract microbiome and allergy. Wang et al. [125] studied the relationship between oral microorganisms and oral environments in polysensitized (sensitized to two to four allergens) atopic individuals; the authors analyzed saliva by multi-omics, reporting that the saliva protein patterns in polysensitized individuals was different from those found in healthy people and concluding that polysensitization was closely related to dysbiosis in the oral microbiota. This indicates that the microbiome manipulation of oral cavity flora could represent a valid alternative for both the prevention and therapeutic treatment of multimorbidity allergies [125].

Moreover, microorganisms can also be the cause of allergies. Many astigmatid (Acari: *Astigmata*) mite species, which infest human habitats producing well-known allergens; additionally, mites are associated with microorganisms that can induce allergic reactions in humans. Tomas Erban and colleagues performed a proteogenomic analysis of 3 *Tyrophagus putrescentiae* populations, a cosmopolitan mite species that can host different bacteria. The authors took advantage of the known *T. putrescentiae* genome and used nanoLC-MS/MS analysis to perform a comprehensive proteomic study. Erban and colleagues were the first to identified WHO/IUIS mite allergens, based on label-free proteomics and *Wolbachia*-specific proteins in mites. These results indicate that individual members of the microbiome can contribute to the development of allergies, as allergens can be derived from bacterial proteins [126].

#### Metaproteomics

Metaproteomics is defined as a technology designed to analyze and identify proteins present in complex environmental microbial communities and it is usually implemented by LC-MS/MS based shotgun method of proteomics [127]. The term was originally published by P. Wilmes and P. L. Bond in 2004 [128]. This -omic technique provides important insights into the microbial activity and signal transduction pathways [129] and providing a snapshot of the metabolic pathways and functions occurring in a microbial community at a particular moment, leading to the understanding of the interactions that occur between the microbiome and the host, both under homeostasis and dysbiosis [130,131,132,133]. Patients suffering from particular GI tract microbiota dysbiosis can display alterations in peptide or protein biomarkers that are not present in their healthy counterparts, hence, a metaproteomic analysis of the gut microbiota could indeed reveal the molecular mechanisms underlying the disease [134]. These analyses can be the base not only for disease diagnosis and prognosis, but also for therapy. Microbial metaproteomics have also been used to analyze different environments, such as soil [135], marine [136] and feces [137] and identify the microbial species they contain. Metaproteomics typical workflow is displayed in Figure 2.

The microbial biomass for microbiome analyses is not only limited, but also contains a variety of contaminants and impurities, hence, usually requires the use of isolation and microbial enrichment methods. The feces of an adult contain approximately 75% of water and 25% of solid material [138], including the gut microbiota [139], while internal body samples of the gut include the mucus layer [140,141], the ileum, cecum and the small and large intestines [127]. The isolation and enrichment methods, necessary for microbiome analyses, include chemical, mechanical and thermal cell lysis techniques as different microorganisms (e.g., Gram-positive and Gram-negative bacteria) require specific conditions for optimal cell lysis [142]. For an effective separation of the microbes, mass spectrometry analyses require the use of high-performance liquid chromatography (HPLC), coupled to rapid, sensitive and accurate mass spectrometry. In fact, metaproteomics successfully utilize hybrid instruments, that combine different types of mass analyzers, such as the Thermo Scientific Q Exactive Hybrid Quadrupole Orbitrap Mass Spectrometer or the Orbitrap Fusion Lumos Tribrid Mass Spectrometer [143]. New technological advances, such as the development of trapped ion mobility spectrometry TOF (timsTOF), constitute a novel way of separating peptide ions, based on their mobility, which results in higher resolution and sensitivity.

The major current challenge in metaproteomics is data analysis, as pointed out by Kleiner. The identification of peptides and proteins and their taxonomic and functional analyses depends on the available information on the genome and protein databases for microorganisms and most of these microbes are currently uncultivable [144]. However, genome and metagenome sequencing collections have dramatically expanded the number of unraveled microbial genomes; in particular, the unified genome catalog includes over 200,000 reference human gut microbiome genomes [145]. The use of metagenomics in metaproteomics is known as metaproteogenomics [146], where proteomics can provide corrections for bad gene predictions and identifying proteinaceous material from specific genome regions that are currently not recognized as coding sequences [147]. Furthermore, novel algorithms and software have been developed for data analysis in different microbiome studies; however, it has the disadvantage that the use of different methodology often results in different data obtained for the same study. The HAPiID (High-Abundance Protein guided metaproteomics Identification) pipeline, a tool to analyze GI tract microbiome data recently described by Stamboulian and colleagues, includes more than 6000 genomes for peptide and protein identification [148]. In addition, over a thousand proteome datasets, obtained from the publicly accessible proteome exchange database, have been analyzed through the HAPiID pipeline. The above results demonstrate that metaproteomic analyses can be applied, both in the study of individual genomes and in the identification of ubiquitous species displaying a variety of phenotypes [149].

In fact, metaproteomics have also proved useful to unravel the relationship between microbiomes and diseases linked to inflammation, such as inflammatory bowel disease (IBD) [141,150], Crohn’s disease (CD) [20,151,152] and allergies [153,154,155]. Maier and colleagues recently reported a technique that used 16S rRNA gene sequencing in combination with metaproteomics and metabolomics to study the effect of dietary resistant starch on the human gut microbiome. The authors analyzed more than 50,000 proteins, found fecal samples and reported an increase in the some species belonging to the *Firmicutes*, hence altering the ratio of this group to Bacteroidetes [156]. In another recent article, Ke and coworkers studied the effect of synbiotic supplements in pathogen-free mice that were fed either a normal or a high-fat diet. The authors found that the synbiotics in diet induced favorably and affected the mouse gut microbiome, restoring homeostasis and reducing the weight gain caused by the high-fat diet [157]. The researchers used a metaproteomic approach that identified more than 11,000 protein groups and 167 KEGG pathways. The Kyoto Encyclopedia of Genes and Genomes (KEGG) is a data base to analyze gene functions. Pan et al. (2020) applied an integrated shotgun metaproteomic approach to study how diet affected microbial protein expression. The authors analyzed fecal samples from subjects consuming either high (refined grain) or low (whole grain) glycemic diets and discovered more than 53,000 unique peptides, 89% of which were of microbial origin, while 11% corresponded to and human peptides. In addition, Pan et al. reported that people consuming refined grain had a significant increase in enzymes involved in the degradation of human mucin [158]. Zhang and coworkers used metaproteomic approaches to study the alterations in the GI tract microbiome associated with pediatric IBD. The authors reported that the microbial proteins related to oxidative stress responses were overexpressed in children suffering from IBD, as compared to their healthy counterparts [141]. In 2015, Kolmeder at al. [159] published a study into the association between obesity and the intestinal microbiota, identifying significant differences in the fecal microbiota from obese and morbidly obese individuals, as compared to people with a healthy weight. The authors analyzed the fecal proteome by, first separation by 1D gel electrophoresis, followed by polypeptide characterization by RP LC–MS/MS; they concluded that the Bacteroidetes species were metabolically more active in the obese individuals [159]. Juste and coworkers conducted a study designed to identify the protein biomarkers from Crohn’s disease (CD) using two-dimensional difference gel electrophoresis (2D-DIGE) and LC-MS/MS. In addition, they applied selected reaction monitoring (SRM) to validate identified biomarkers and 16S rRNA gene sequencing to analyze the structure and functions of the GI tract ecosystem. The results revealed that people suffering from CD had elevated levels of *Bacteroides* proteins and reduced quantities of *Firmicutes* and *Prevotella* polypeptides, as compared to healthy individuals; while specific proteins for CD, that can be used as biomarkers for the disease, perform a role in bacterial colonization of the mucus layers, by helping the microorganism breach the host intestinal barrier and invade the GI tract mucosa [151]. Robert and colleagues reported a negative association of *Faecalibacterium* and a positive association of *Escherichia* with calprotectin in Crohn’s disease patients. In conclusion, all these metaproteomic studies clearly stablished a GI tract dysbiosis as the origin of CD and identifying alterations in urease activity and amino acid metabolism, promoting a chronic inflammation state [20].

Kingkaw et al. [154], in a metaproteomic study conducted in Thailand, identified reporter proteins displaying metabolic alterations in the microbiome of infants suffering from atopic dermatitis, as compared to their healthy counterparts. Fecal samples were analyzed by liquid chromatography-tandem mass spectrometry and eight proteins were determined as biomarkers for the atopic dermatitis disease; although the eight biomarkers were involved in metabolic processes, the most interesting proteins were only expressed in individuals suffering from the disease; these include triosephosphate isomerase (TPI), produced by the bifidobacterium *Alloscardovia* and dimethylmenaquinone methyltransferase (DMM) from *Bacteroides* [153].

The GI tract mycobiome (fungi) in mammalians is estimated to represent less than 0.1% of the gut ecosystem, but it performs an important role in host disease development [160]. Mok and colleagues studied the fungal microbiome of 9- to 12-month-old infants with atopic dermatitis, organized into two groups (recovered or still undergoing) according to their symptoms. They used both metagenomic and metaproteomic approaches (LC–MS/MS), to conclude that the diversity of the mycobiome was higher in the cohort still suffering from the disease. The infants undergoing GI dysbiosis displayed an increase in *Rhodotorula* sp. and a reduction in the Ascomycota/Basidiomycota ratio, while *Wickerhamomyces* and *Kodamaea* species were significantly increased in the healthy group. Interestingly, microorganisms belonging to either the genus *Acremonium* or *Rhizopus* were enhanced in the recovered group. In addition, the authors identified five fungi as biomarkers from each sample group and used a metaproteomic approach to determine that the sick cohort had a higher abundance of fungal proteins, with *Rhodotorula* sp. as the main producer. In addition, this common environmental yeast, generates two unique proteins, RAN-binding protein 1 and glycerol kinase, that are only present in infants with atopic dermatitis, suggesting that they are involved in the development of such syndrome [155].

Petersen and coworkers used gnotobiotic mice, colonized with a specific consortium of microorganisms, including twelve bacterial species, five fungal species or both, to study the effect of substances, such as antibiotics or antifungals that alter the GI tract microbiome on mouse pups. The authors analyzed feces by Label-Free Quantitative (LFQ) proteomics, while samples from the small intestine of the mice underwent labelling with Tandem Mass Tag (TMT). The authors found that the antimicrobial treatments produced lasting effects of both bacteria and fungi, indicating that they effect extended to the whole GI tract microbiome, but fungal colonization appeared to have the most drastic effect on the pups. The presence of certain fungi in the GI, in the early stages of development, produced changes in host proteins involved, not only in innate immunity, but also in metabolism. The strong impact of some fungal species on the host intestinal proteome is not surprising as some fungal proteins were reported to circulate in extracellular vesicles and exhibit immunomodulation properties [160].

## 4. Future Perspectives

The fact that some microorganisms, and particular bacterial metabolites, can induce the production and secretion of immunoglobulins [10], it is not surprising that modifications in the GI tract microbiota can result in immunomodulatory alterations in the intestinal mucosa, which originate FA. Hence, restoring homeostasis in the GI tract microbiota is a means to prevent or even treat FA. One such treatment is healthy fecal microbiota transplantation, as it contains beneficial commensal bacteria that can reverse GI tract inflammation and restore intestinal health [5,25]. The current exponential increase in FA, particularly in developed countries, requires the development of improved methods for the prevention, diagnosis and treatment of these GI tract dysbioses. However, to achieve this, a better understanding of the alterations in the gut microbiota that result in dysbiosis is essential.

The use of proteomics, to study both the allergenic components of particular food items and the process by which they cause FA, is currently producing an array of useful data, increasing the available knowledge. However, although more sequence information on allergens is available, the structure and, in particular, the reactivity of food allergens remains largely unknown [43]. The ideal future treatment of diseases would involve the use of multi-omics, in order to predict not only, the patient’s physiological response to a particular disease, but also to personalize the medical treatment [161]. The immunology of the future would require, not only the identification, but also the absolute quantitation of allergens present in a particular environment, and this would require the development of novel techniques, currently still lacking. Although we are currently experiencing a resurgence in this field, spearheaded by technological advances, such as High-Resolution Mass Spectrometry, that permits molecular allergy sciences to study the mechanisms that affect the allergenicity of food by matrix interferences and to unravel the structural elements involved in the mechanisms of sensitization and in allergic reactions of novel diagnostic techniques, this will undoubtedly result in more efficient therapeutic approaches [162].

An additional advantage of metaproteomic approaches is that they can determine taxonomy, function and metabolic pathways present in the GI tract microbiome, and identifying all the proteins that are synthesized by microbial communities at a particular time [127]. The main challenge in metaproteomics is to develop better ways to analyze the data obtained from high-resolution MS/MS, as the current algorithms are only designed to identify single microbial species, while GI tract microbiomes are rather complex, both in size and the number of organisms present [142].

## Figures and Tables

**Figure 1 ijms-24-02234-f001:**
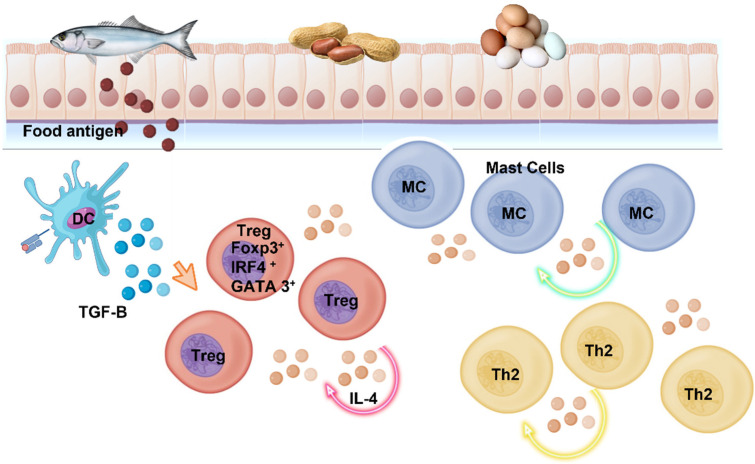
In patients suffering from FA, the Tregs in the gut mucosa can acquire a pathogenic, TH2 cell-like, phenotype. These pathogenic Tregs display high levels of GATA-3 and increased IL-4 production, resulting in the accumulation of dysfunctional antigen-specific Tregs, which fail to control effector TH2 and mast cell responses, promoting allergic disease. Th2 cells (yellow), Tregs cells (red), mast cells (blue).

**Figure 2 ijms-24-02234-f002:**
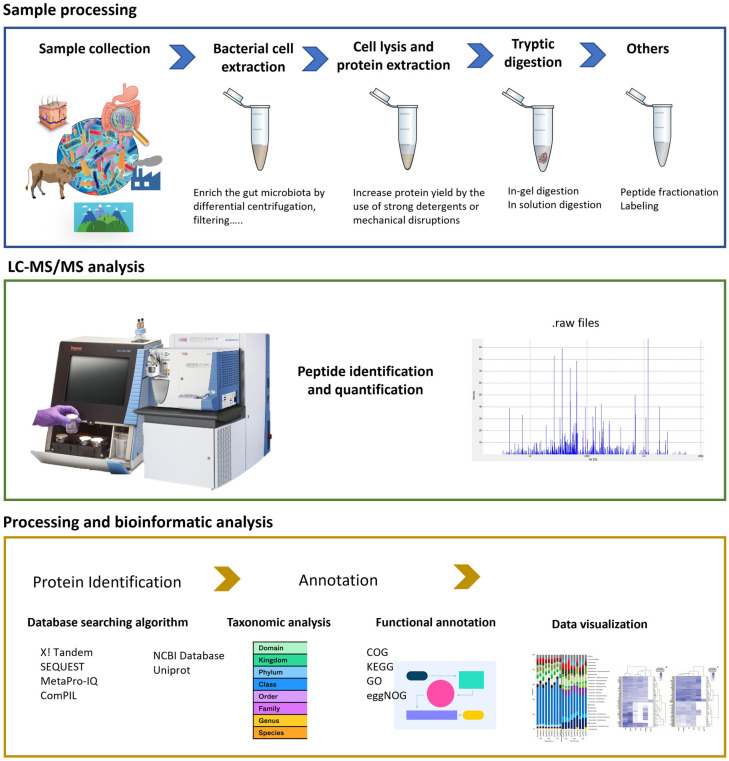
Shotgun metaproteomics approach to identify microbial proteins in human and ambient samples.

**Table 1 ijms-24-02234-t001:** Relevant alterations in the composition of GI tract microbiota in the presence of FA.

Types of Food Allergy	Association with Food Allergy	Reference
Cow’s milk	↓Clostridia, Firmicutes	[15]
Cow’s milk, egg, wheat, soy, nuts	↓*Citrobacter*, *Oscillospira*, *Lactococcus*, *Dorea*	[16]
Cow’s milk, egg, peanut	↑Enterobacteriaceae ↓Bacteroidaceae	[17]
Peanut	↓Clostridiales ↑Bacteroidales	[18]
Egg, wheat, soybean, sesame, cow’s milk, peanut, shrimp, crab	↓*Dorea*, *Akkermansia* ↑*Veillonella*	[19]
Cow’s milk, egg, wheat, nut, peanuts, fish, shrimp, soybeans	↓Bacteroidetes, Proteobacteria, Actinobacteria ↑Firmicutes	[20]
Egg	↑Lachnospiraceae, Streptococcaceae, Leuconostocaceae	[21]
Cow’s milk	↑Lactobacillaceae ↓Bifdobacteriaceae, Ruminococcaceae	[22]
Cow’s milk	↓Coriobacteriaceae	[23]
Cow’s milk	↑Bacteroides, *Alistipes*	[11]
Tree nuts, fish, milk, egg, sesame, soy	↑*Oscillibacter valericigenes*, *Lachnoclostridium bolteae*, *Faecalibacterium* sp.	[24]

## Data Availability

Not applicable.

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
