# Peer review of "Gut Microbiome Proteomics in Food Allergies"

_ijms, 2023, doi:10.3390/ijms24032234_

Round 1

Reviewer 1 Report

The review by Abril and colleagues focus on the potential role of the gut microbiome and its metabolites in food allergies. It discusses about the use of metaproteomics technologies to study both the allergenic components of food and the process by which they cause food allergies and the GI microbiome phenotypical and functional composition in allergic subjects. These could represent the base, for new disease diagnoses, prognoses, and therapeutic approaches.  The review is interesting and well-detailed.

I have some suggestions for the authors regarding the form:

Page 1, line 44: replace “bacterial” with bacteria

Page 2, line 52 and along the manuscript: use the italics only for genus and species taxa (not for orders like Clostridiales and Bacteroidetes).

Page 2, line 82: Parabacteroides is a genus, not a family taxon

Page 2, line 84 and 92: the plural of "taxon" is "TAXA", not taxons. Please revise the test.

Page 2, line 94: add the reference number 10.

Page 2-3, lines 97-102. The study of Bao and colleagues is the same as discussed in the previous lines (reference number 10). Revise the paragraphs, since it seems a different finding. Moreover, the study of Bao et al, which performed the 16s RNA sequence analysis, cannot accurately infer the microbial species. So the results presented in terms of species must be evaluated with criticism.

In many cases the abbreviation GI (gastrointestinal) should be followed by an object, for example, “GI tract” (i.e. page 1, line 34; page 2 lines 55 and 93; page 3 line 114, ecc… look along the manuscript).

Page 3, lines 123-124: Please revised the sentence “the plasma cells present in the intestinal lamina propria (the layer 123 of connective tissue located under the mucosal epithelium), upon differentiation into B cells”. Plasma cells are derived FROM B cells.

Page 5 line 176: I think the authors should add Response at the end of the sentence (…humoral immune.)

Page 6, line 233: authors started to use the abbreviation FA instead of food allergies. Please, start using the abbreviation from the first use of the term and explain it.

Page 8, lines 331-332: “it also successfully recognized lymphoma neoantigens 331 [38], neoantigens are novel, altered, proteins that originate from cancer cell-specific DNA mutations“. Please revise this sentence without repeating “neoantigens” term (for example using brackets for neoantigens explanation).

Page 8 line 369: please revise “one such example an example…”

Page 12, line 526: Please correct the term “symdrom”

Author Response

Thank you very much for your advice, we have revised the manuscript and carefully proof-read it. All suggestions have been accepted by the authors:

Page 1, line 44: replace “bacterial” with bacteria Done

Page 2, line 52 and along the manuscript: use the italics only for genus and species taxa (not for orders like Clostridiales and Bacteroidetes). Done

Page 2, line 82: Parabacteroides is a genus, not a family taxon. Corrected

Page 2, line 84 and 92: the plural of "taxon" is "TAXA", not taxons. Please revise the test. Corrected

Page 2, line 94: add the reference number 10. Added.

Page 2-3, lines 97-102. The study of Bao and colleagues is the same as discussed in the previous lines (reference number 10). Revise the paragraphs, since it seems a different finding. Moreover, the study of Bao et al, which performed the 16s RNA sequence analysis, cannot accurately infer the microbial species. So the results presented in terms of species must be evaluated with criticism. Reference removed from text 11.

In many cases the abbreviation GI (gastrointestinal) should be followed by an object, for example, “GI tract” (i.e. page 1, line 34; page 2 lines 55 and 93; page 3 line 114, ecc… look along the manuscript). Manuscript corrected.

Page 3, lines 123-124: Please revised the sentence “the plasma cells present in the intestinal lamina propria (the layer 123 of connective tissue located under the mucosal epithelium), upon differentiation into B cells”. Plasma cells are derived FROM B cells. Corrected.

Page 5 line 176: I think the authors should add Response at the end of the sentence (…humoral immune.) Added.

Page 6, line 233: authors started to use the abbreviation FA instead of food allergies. Please, start using the abbreviation from the first use of the term and explain it. Done

Page 8, lines 331-332: “it also successfully recognized lymphoma neoantigens 331 [38], neoantigens are novel, altered, proteins that originate from cancer cell-specific DNA mutations“. Please revise this sentence without repeating “neoantigens” term (for example using brackets for neoantigens explanation). Modified to: “it also successfully recognized lymphoma neoantigens (modified proteins generated by cancer-specific DNA mutations) 331 [38]”. Done

Page 8 line 369: please revise “one such example an example…” Corrected

Page 12, line 526: Please correct the term “symdrom” Corrected to syndrome.

Reviewer 2 Report

The authors discuss a relevant and interesting topic, and provide an in depth understanding of the molecular / microbiological mechanisms underlying food allergy. The paper may be suitable for publication, provided that the following issues are being resolved:

- The authors make a lot of use of colons, commas, and semicolons, which makes the text difficult to read. Please make appropriate usage of this throughout the text and correct. Also, sentences are often very long. Please improve readability by splitting long sentences into two.

- The fact that allergies may depend on geographical location, like cow milk allergy / lactose intolerance, should be discussed in detail in light of possible gut microbial differences between populations originating from different geographical locations. On top of that, it is well known that there exists both large interindividual and intraindividual variation in gut microbiota composition (please refer to:  Larsen, O. F. A., E. Claassen, and Robert Jan Brummer. "On the importance of intraindividual variation in nutritional research." Beneficial microbes 11.6 (2020): 511-517.). This interindividual variation complicates the identification of microbial signatures associating with allergies, and as such may also preclude the possible usage of "one-size-fits-all" interventional modalities. Intraindividual variation complicates the diagnosis of allergies and the underlying gut microbial signatures, as they can vary in time due to fluctuating external factors. The authors should discuss this in their paper as well.

Author Response

Thank you very much for your advice, we have revised the manuscript and carefully proof-read it.

Following the referee's suggestions we have improved readability and the English language and style were re-edited.

Following the referee's suggestions we have cited the article in the manuscript and we have discussed it in the text ( from page 1, line 46 to page 2 line 55).
